# Effects of Early Retirement Policy Changes on Working until Retirement: Natural Experiment

**DOI:** 10.3390/ijerph16203895

**Published:** 2019-10-14

**Authors:** Cécile R.L. Boot, Micky Scharn, Allard J. van der Beek, Lars L. Andersen, Chris T.M. Elbers, Maarten Lindeboom

**Affiliations:** 1Department of Public and Occupational Health, Amsterdam Public Health Research Institute, Amsterdam UMC, VU University, 1081 BT Amsterdam, The Netherlands; m.scharn@amsterdamumc.nl (M.S.); A.vanderbeek@amsterdamumc.nl (A.J.v.d.B.); 2National Research Centre for the Working Environment, 2100 Copenhagen, Denmark; lla@nrcwe.dk; 3Department of Economics, VU University, 1081 HV Amsterdam, The NetherlandsM.lindeboom@vu.nl (M.L.)

**Keywords:** retirement, Netherlands, older workers, policy, transitions, longitudinal

## Abstract

Many European countries have implemented pension reforms to increase the statutory retirement age with the aim of increasing labor supply. However, not all older workers may be able or want to work to a very high age. Using a nation-wide register data of labor market transitions, we investigated in this natural experiment the effect of an unexpected change in the Dutch pension system on labor market behaviors of older workers. Specifically, we analyzed transitions in labor market positions over a 5-year period in two nation-wide Dutch cohorts of employees aged 60 years until they reached the retirement age (*n* = 23,703). We compared transitions between the group that was still entitled to receive early retirement benefits to a group that was no longer entitled to receive early retirement benefits. Results showed that the pension reform was effective in prolonging work participation until the statutory retirement age (82% vs. 61% at age 64), but also to a larger proportion of unemployment benefits in the 1950 cohort (2.0–4.2%) compared to the 1949 cohort (1.4–3.2%). Thus, while ambitious pension reforms can benefit labor supply, the adverse effects should be considered, especially because other studies have shown a link between unemployment and poor health.

## 1. Introduction

Modern industrialized countries have well-developed pension systems that provide income support after retirement. These pension systems also act upon financial incentives that influence retirement behavior. More specifically, retirement benefits are lower when individuals decide to retire at early ages, and they tend to increase when a worker delays retirement. Thus, the individual worker considering retirement is faced with the trade-off between accepting lower benefits and more years in retirement versus higher benefits, but fewer years in retirement. Economic factors play a role in retirement considerations of older workers and one in every three to four workers would consider staying longer if it would pay better off economically [1]. Previous research has shown that the differences in the design of country-specific pension systems explains most of the cross-country variation in retirement [2]. Consequently, faced with population aging and declining fertility rates, many countries have started to implement pension reforms that are aimed at increasing the labor supply of older workers by either abolishing or reducing the relative benefits of early retirement. These reforms appear to have been effective as labor force participation has increased in all countries that have implemented such reforms. While aggregate effects of financial incentives in pensions schemes have been studied extensively (see e.g., [3]), much less is known about the dynamic labor market responses of specific groups of workers nearing retirement. This is what this paper is about. More specifically, we exploit an exogenous and unexpected change in the Dutch pension system that substantially changed the incentive structure of a pension scheme to causally assess its effect on labor market behavior of workers nearing retirement ages. The reform acted as a near-perfect natural experiment as no other major reforms were performed around this time. 

In short, in the summer of 2005 it was announced that as of January 2006, the pension rights of those born in 1950 (or later) were substantially reduced, while the pension rights of those born in 1949 (or earlier) were unaffected. The group born in 1950 or later either had to work another 13 months to obtain the same pension rights, or would have to accept a pension of 64% instead of 70% of their gross wages if they wanted to retire at age 62 years and 3 months. We could therefore apply a sharp regression discontinuity design. It is expected that those who are affected by the reform will postpone retirement, but the reform may also affect transitions to other non-work states of the labor market, such as Work Disability Insurance (DI) or Unemployment Insurance (UI). DI may become more common when ‘treated’ individuals reaching the retirement age are not able to work the extra 13 months because of health problems. Furthermore, low productive workers are at risk for unemployment for a longer period of time, potentially leading to more UI. Indeed, previous work by Riphahn [4] and Kerkhofs and collagues [5] has shown that DI and UI schemes interact with pension schemes. Therefore, a complete evaluation of the extent to which pension reforms influence retirement also requires insights into the effects of these reforms on transitions into other states of the labor market. 

Previous studies have in other contexts investigated financial incentives on retirement decisions. This paper addresses the more general issue of the effectiveness of pension reforms and has a combination of several factors that makes this paper distinct from previous contributions. First, the clear and discontinuous change in the pension system allowed us to implement a sharp discontinuity design. Second, we were able to use population-wide administrative data of a narrowly defined homogenous group of workers that only differ in their treatment status. This design is as close as it can get to an experimental set-up and therefore substantially reduces the risk of confounding [6]. Third, with simple and flexible methods that require no distributional assumptions we assessed the full impact of the reform on all labor market transitions. This provides a better insight into the full spectrum of labor market outcomes in the latest stages of working life of men and women. We show that pension reforms have the intended effect, as they generally lead to delayed retirement. However, a part of the intended policy effects leak away via outflow to other non-work states, such as unemployment and work disability. 

## 2. Materials and Methods

### 2.1. Data

This study made use of linked register data of Statistics Netherlands, covering the entire population. Two datasets were selected: data from the income tax (SECMBUS) and data from the municipality (labelled GBAPERSOONTAB by Statistics Netherlands). SECMBUS provides information of individual tax records from 1999–2016. From this database we used the information about source(s) of income to categorize individuals according to their labor market position. As people only pay tax if they have an income, the register is highly valid in relation to labor market status. The dataset GBAPERSOONTAB consists of demographic characteristics of individuals who were registered in the Dutch municipality register since 1995. Both datasets were linked at the individual level based on a unique identifier. 

To classify individuals according to their labor market position we used the labor market position on January 1 of each year. This implies that we did not take into account multiple changes between labor market positions within one year. 

From the 2010 administrative files, we selected all 76,490 individuals who were born between 1 November 1949 and 28 February 1950. After deleting incomplete records, we selected a random sample of 80% for analyses to reduce the likelihood for identification of persons. This left us with 56,225 individuals, of which 23,703 were employed wage earners and 5569 self-employed. Our main analyses focused on employed wage earners, and the group of self-employed workers was analyzed for the purpose of testing external validity (Figure 1).

### 2.2. Design

We applied a sharp regression discontinuity design. The cohort born in 1949 served as a control group and the cohort born in 1950 as a treatment group to investigate the effects a pension policy reform.

The unaffected cohort (born in 1949 or earlier) could retire at age 62 years and 3 months at 70% of their gross wages. For cohorts born in 1950 or later, the gross replacement rate dropped to 64% if they wanted to retire at age 62 years and 3 months. To obtain the same benefits as their older counterparts, they had to work an additional 13 months. 

From this database we selected employed wage earners in 2010, who only differed in their ‘treatment status’: those who were not affected by the reform, born in November–December 1949 (the controls), and those affected by the reform, born in January–February 1950 (the treated). 

Besides this reform, no other institutional changes differentially affected the November–December 1949 and the January–February 1950 cohorts in 2006 (i.e., at the same time as the 2006 pension reform in 2006, the Dutch government introduced the so-called Life Course Savings Scheme (Levensloopregeling)). This tax-facilitated savings program enabled all workers (those treated by the pension reform as well as those who are unaffected) to privately save at lower costs. Workers could use these savings to self-finance earlier retirement. 

Cohort 1949 (control group): individuals born in November or December 1949 and registered as being employed at the time they reached the age of 60 years in November or December 2009. This cohort included workers who were not subject to the policy reform of January 2006.

Cohort 1950 (treatment group): individuals born in January or February 1950 and registered as being employed at the time they reached the age of 60 years in January or February 2010. This group was the first cohort for whom retirement options was reduced substantially. More specifically, their slightly older counterparts born in November–December 1949 (the controls) could retire at age 62 years and 3 months, with a pension benefit of 70% of the average income in preceding 5 years. Those born in January–February 1950 (the ‘treated’) faced a 6% point drop in their pension benefit if they retired at age 62 and 3 months, or they had to work 13 months longer to get the benefit level of 70%. 

We compared both cohorts until the statutory retirement age of 65 years and 3 months in 2015.

For internal validity of the research design, it was crucial that workers born in the 1950 cohort are aware of the consequences of the new pension system for their individual situation. Previous work by Lindeboom and Montizaan [7] looked at the retirement expectations of public sector workers born in 1949 and 1950. It was found that workers do adjust their expectations in accordance with the incentives implied by the reform. Using this design, we followed both groups over time up to the statutory retirement age (65 years and 3 months) and used flexible non-parametric (Kaplan–Meier) estimates of all of their labor market transitions up to the mandatory retirement age. In this way, we causally assessed the full effect of the reform on the full labor market dynamics of older workers between age 60 and the statutory retirement age of 65 years and 3 months. 

### 2.3. Labor Market Positions

For our main analyses on employed wage earners, we distinguished the following labor market positions: Employment: individuals receiving earnings from an employer in the Netherlands or abroad. 

Early retirement: individuals registered as ‘no status’ as only labor market position following an episode of paid work (i.e., either in self-employment or employment). This includes self-financed periods of retirement preceding transitions into official retirement schemes. 

Work disabled: individuals receiving work disability benefits.

Unemployed: individuals receiving unemployment benefits. 

Other: individuals receiving means-tested Social Assistance benefits, and any other benefits, such as partial work disability benefits for unemployed (older) workers (following employment), benefits for artists, and work disability benefits for young individuals. Moreover, as we focused on employment, employed wage earners who transitioned into self-employment were also categorized into this category. 

Self-employment was defined as receiving profit from entrepreneurship, or from other paid activities, or if they were registered as director or large shareholder as main source of income, rather than receiving earnings from an employer. 

More than one labor market position: individuals with more than one main source of income were categorized into one category based on the following hierarchy, based on highest to lowest societal costs: work disability > unemployment > other benefits > employment > early retirement. 

Lost to follow-up: individuals who no longer paid taxes in the Netherlands, e.g., following death or a move to another country after 2010, and were no longer present in the registers.

Table 1 describes the data of *n* = 23,703 male and female employees and 5569 male and female self-employed workers from the 1949 and 1950 cohorts. Note that at the start of our observation period in January 2010, 50% of the males and 34% of the females were still at work. The percentage of employees in the 1949 cohort was slightly lower compared to the 1950 cohort; the percentage of workers with a Dutch background was similar for both cohorts. Non-Dutch background was slightly more common for employees (13–14%) compared to self-employed workers (10–12%).

### 2.4. Analyses

The pension reform affected all employees born in or after 1950. Therefore, in our main analyses, we selected employees on 1 January 2010 and yearly followed their labor market position until they reached the statutory retirement age (65 years and 3 months). We did this both for the controls (the 1949 cohort) and the treated (the 1950 cohort), separately for males and females. We also stratified with respect to Dutch/non-Dutch background; this did not provide additional insights (data not shown). For each labor market transition, we recorded the state of destination and calculated discrete time equivalents of the Kaplan–Meier estimate of the transition rate out of work to state *j* as:(1)$$\theta_{t}^{j}=\frac{\sum_{i}I_{it}^{j}}{N_{t}},$$
where *t* indexes time, *i* the individual, and *j* the state of destination, *j* ∈{Employed, Early retired, Unemployed, Disabled, Other}. $I_{it}^{j}$ is an indicator function that equals 1 for an individual that makes an exit to state *j* and *N_t_* the number of employed at time *t*. With the different transition rates out of employment, we had a competing risk setting where the total risk out of employment (θ_t_) equaled the sum of the risks (transition rates) to the other states *j, j≠Employed*:$$\theta_{t}=\sum_{j\neq Employed}\theta_{t}^{j}.$$

To examine whether there were spill overs from the pension reform to other labor market states, we compared exit from employment to early retirement, work disability, and unemployment between the controls and the treated. 

### 2.5. External Validity: Analyses for Self-Employed Workers

Since the reform did not affect self-employed workers, we compared the results for wage earners with the results for self-employed to provide insights into how generic societal trends related to work participation may have influenced labor market transitions independent of the change in the pension system. More specifically, similar to the analyses for employees, we selected a random sample of 80% of the self-employed workers in 2010, followed these up to age 65 and 3 months and recorded their transitions. If we found substantial differences between the 1949 and the 1950 cohort, then this was indicative of other structural changes besides the pension reform affecting labor market behavior of older workers.

## 3. Results

### 3.1. Continuing Employment until the Statutory Retirement Age

Table 2a shows that, for males, already in the first year after 2010 employment rates dropped substantially for both cohorts. Of the 1949 born who were at work in January 2010, 85% was still at work one year later and this drops to 61% in the final year prior to the statutory retirement age (2014–2015). For the treated, the employment rates are substantially higher: ranging from 90% in the first year to 82% in 2014–2015. The same pattern emerges for females. Note that the exit rate out of work is higher for females. Only 34% of the females were still employed in January 2010 and these might be women with stronger attachment to the labor market than other (non-working) females and their male counterparts. A comparison of columns ‘Working’ of Table 2a,b thus shows that the reform was effective in prolonging working life of wage earners. 

Figure 2a,b graphically depicts the drop in employment rates of both cohorts for both genders. Note that the tables represent all employed wage earners per year, and allow employed wage earners to return to employment following an episode of e.g., unemployment, whereas for Figure 2a,b, employed wage earners are censored once they leave employment. The hazard ratios represent the likelihood of working until the statutory retirement age in the 1950 cohort (intervention) compared to the 1949 cohort (control). Male and female workers of the 1950 cohorts were significantly more likely to work until the statutory retirement age compared to the 1949 cohorts, although the effect was smaller in females than in males (Hazard Ratio (HR) males 1.186; 95% Confidence Interval (CI): 1.166–1.205 / HR females 1.088; 95% CI 1.067–1.110) (Figure 2a,b).

### 3.2. Exits from Employment to Other Labor Market States

Table 2a,b shows the exit rates to different labor market states for males and females, respectively. The larger part of the transitions out of work are to early retirement, in line with the incentives of the pension reform. For the 1949 cohort (the controls) the transition rate is highest at around age 62–63, the age at which the controls become eligible for early retirement benefits. For the 1950 cohort, the transition rates to early retirement are much lower (often only half of the transition rate of the controls), but there does not appear to be a clear spike at age 63–64 as one would have expected a priori. This is reflected in the high fraction of wage earners born in 1950 who continue to work up until the mandatory retirement age. A comparison of unemployment between the 1949 and 1950 cohorts shows that the transition rates to unemployment are also elevated for the treated cohort. Figure 3a,b gives a graphical representation of how the employed wage earners in 2010 prolong working from 2010 to 2015. 

Overall, exit from employment through early retirement was less common in the 1950 cohorts (4.1–12.1%) compared to the 1949 cohorts (7.8–27%). Exit through unemployment (2.0–4.2%) was more common in the 1950 cohorts compared to the 1949 cohorts (1.4–3.2%). For work disability, this was found for the first two years, but this difference was not found in the final years before reaching the statutory retirement age (2014/2015).

### 3.3. Self-Employed Workers

Compared to the employed wage earners, the proportion of self-employed workers continuing self-employment until the statutory retirement age was more stable over time as can be seen in Table 3 and Figure 4. Until 1 year before reaching the statutory retirement age, around 90% of male self-employed workers continued self-employment until the statutory retirement age. For females, 82–85% of the self-employed workers continued working until the statutory retirement age. Importantly, for both genders we found no significant differences between the 1949 and 1950 cohorts with regards to working until the statutory retirement age (HR males: 1.015; 95% CI 0.985–1.046 / HR females: 1.023; 95% CI 0.982–1.065) (Figure 4). Note that the exit rate out of work is higher for females. This might be due to selection effects. Only 34% of the females were still employed on January 2010 and these might be women with stronger attachment to the labor market than other (non-working) females and their male counterparts.

## 4. Discussion

Our results show that the pension reform from 2010 had a major impact on older employees born in 1950. The majority of employees affected by the reform (born in January/February 1950) continued employment until a higher age compared to their slightly older counterparts (born in November/December 1949) who were not affected by the reform. However, a larger part of the older employees affected by the reform did not continue working, but left the workforce through social benefits by making use of unemployment insurance or, to a lesser extent, work disability insurance. We conclude that reducing financial incentives for early retirement indeed explains the increase in average retirement age at population level. However, this increase also comes with some additional costs to be paid by society mainly, following from unemployment, both in men and women. 

The effects of the pension reforms from 2010 on prolonged work participation and exit from work through other routes can be explained both from an individual perspective and from a societal perspective. From the individual perspective, it is likely that the pension reform has led to a selection effect in the population of older workers since only workers with a higher level of income or more assets may have been able to afford exit from work through early retirement. This is in line with findings by Lindeboom and Montizaan [7] who showed that, in particular, people with a higher level of income participated in a life course savings program (a tax facilitated savings program that was introduced at the time of the pension reform). They argue that, therefore, those with a higher level of income could more easily counter the reduction in pension wealth induced by the pension reform. This leaves the subgroup with a lower level of income to work until a higher age [8]. Those with a lower level of income are likely to have worse working conditions and poorer health. Within the population of older workers, poor health is common. For example, the prevalence of chronic disease increases with age, 46% of workers aged 55–64 years reported at least one chronic disease in 2018 [9]. It is widely known that poor health and chronic diseases are associated with early exit from work (e.g., in [10]). Early retirement has been shown to be a good alternative for workers with chronic disease when continuation of work is no longer possible [11]. Removing the possibility for early retirement may leave a large group of workers stuck in the labor market with poor health but without the financial possibility to withdraw. Using unemployment—and thereby getting unemployment benefits—may be the only realistic option for this group of workers until they reach statutory retirement age. Our findings on prolonged work participation are in line with a recent study on older workers with chronic disease that have prolonged their working careers by an additional 4 months over the past 20 years [11]. This selection effect where workers with poor health cannot afford early retirement, and as a result continue working until a higher age, may also explain the increased mortality reported in previous studies for groups retiring at a higher age [12]. Moreover, once retired, favorable effects of retirement on health are mainly present in higher socioeconomic groups as was shown in a recent review [13,14]. It is questionable if the vulnerable but large subgroup of the total group of older workers, i.e., those with poor health, should be encouraged to delay their retirement by prolonged work participation until a higher age as this may increase existing socio-economic health inequalities.

From a societal perspective, the increase in unemployment and work disability insurance leads to societal costs. Although this study does not take into account costs associated with alternative exit routes from work, it is evident that policy changes that seem effective in reducing exit through early retirement, may lead to extra costs resulting from an increase of social benefits. This redistribution of costs should be taken into account when deciding about policy measures related to prolonging work participation. Overall, prolonging work until a higher age is becoming more common nowadays, whereas early retirement was the standard in previous years, also for groups who were physically and mentally capable to continue working until the statutory retirement age. In line with these societal changes, future studies could focus on building a business case from the societal perspective for prolonged work participation, and such studies should take into account costs associated with alternative exit routes. 

It should be noted that for the 1950 cohort, the transition rates to early retirement are much lower (often only half of the transition rate of the controls), but there does not appear to be a clear spike at age 63–64 as one would have expected a priori. To understand this one has to realize that the change in the pension system did not simply shift the incentive structure to the right of the age distribution, but that in general continued work at later ages (beyond age 63) is stimulated with large accruals in pension wealth with continued work at later ages.

We stratified our analyses for men and women, given the differences in work participation. A striking finding was that women were more likely to continue employment until the retirement age compared to men. There are two explanations for this. First, the cohort of women working is a highly selected group as the employment rates of women in the 70s and 80s were much lower compared to those of men in the Netherlands. This group of women working might be the first group to have a working career, as only 42% of the females were involved in paid work compared to >60% of males. Another explanation may be that women had lower pension savings resulting from a lower income or a non-continuous working career, as in the 70s and 80s it was quite common to quit working to take care of children. Part of the group of women in this study might have re-entered the labor force once their children were grown up. 

Our sensitivity analyses on self-employed workers strengthens the principal findings, i.e., except for the last year before reaching the statutory retirement age, no major differences were found between the 1949 and 1950 cohort. The increase in exit from self-employment in the last year before retirement may be explained by pension savings from a previous job in employment. Self-employment has been used as a strategy to prevent unemployment as between 2010 and 2015, there was an economic crisis in the Netherlands, during which many organizations were restructured, leading to exit from work to unemployment [15]. As it was (and is) difficult for older workers to find a new job, self-employment was a way out of unemployment providing income from work [15]. This might explain why self-employed workers from the 1950 cohort show a small effect from the pension reform. However, since the pension income is highly dependent on the final years of the working career, the effects of the reform on this group are likely to be much smaller, as was clearly demonstrated in this study.

### Methodological Considerations

A major strength of this study is the sharp discontinuity design that was applied on complete data of all workers in the Netherlands born from October 1949 until February 1950, between 2010 and 2015. Since the pension reform was announced in 2005, employees had very little time to take financial measures to counteract the additional 13 months that had to be worked to reach the same amount of pension benefits. 

One limitation of the data we used is that we had to rely on source of income. This implies that early retirement is not a separate category, as this depends on the origin of the early retirement benefits. These may originate from insurance, but may also be personal savings. In the present study, we operationalized early retirement as no status with regards to source of income. Given the fact that we included employees only, the group that no longer received income from employment, self-employment, or any insurance or benefit was considered to be retired. This might have caused bias, as this might also include some persons who are unemployed but are not or no longer entitled to unemployment benefits as they had not worked sufficiently in the past. However, we do not expect that these groups are very large, since we only included employees aged 60 years in this study.

Another limitation of this study is that we did not include all labor market transitions, but focused on the labor market position of January of each year. This might have caused an underestimation of the effect, since prolongation of employment of less than 12 months was only partially taken into account. E.g., two workers in the 1950 cohort who were entitled for early retirement starting in February 2013, of whom one decided to exit work in February, and the other decided to prolong employment until December 2013, would both be listed as employees in 2013. In addition, when individuals had more than one labor market position, we chose a hierarchy based on societal costs to avoid an underestimation of additional costs for society, therewith accepting an underestimation of the proportion of workers prolonging work. Finally, we only had access to a limited set of observed characteristics. Therefore, we could not control or further disaggregate the results by other categories. Nevertheless, if one is interested in effectiveness of the reform on prolonged work participation, the data presented in this study are sufficient. In future, for more detailed studies about the heterogeneous impact of the reform on e.g., health, more detailed information is required. 

## 5. Conclusions

Using complete data of the Dutch population of workers born between October 1949 and February 1950 from 2010 to 2015, we were able to demonstrate that the 2005 pension reform caused a significant prolongation of work participation. However, not everybody in the group of workers that could no longer exit work through early retirement continued working until the statutory retirement age, since the proportion of workers receiving social benefits from unemployment insurance or work disability insurance increased. 

The pension reform has likely affected an already vulnerable subgroup of older workers with a lower level of income, poorer health, and worse working conditions. Future research is needed to investigate if prolonging work participation until a higher age has negative effects on health in the longer term, and if policies with short-term beneficial effects on work participation of older workers may lead to higher societal costs in the longer term, such as costs related to unhealthy ageing. 

## Figures and Tables

**Figure 1 ijerph-16-03895-f001:**
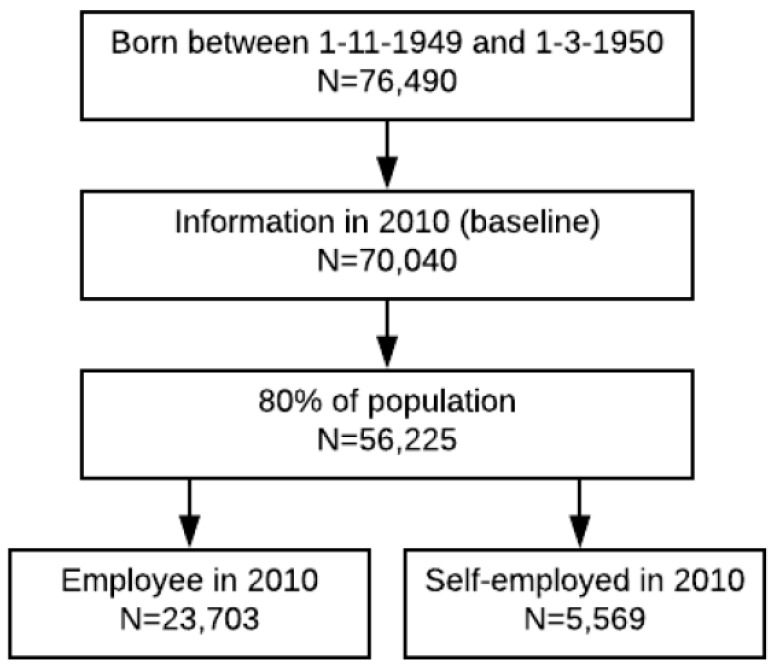
Flow diagram of the study population.

**Figure 2 ijerph-16-03895-f002:**
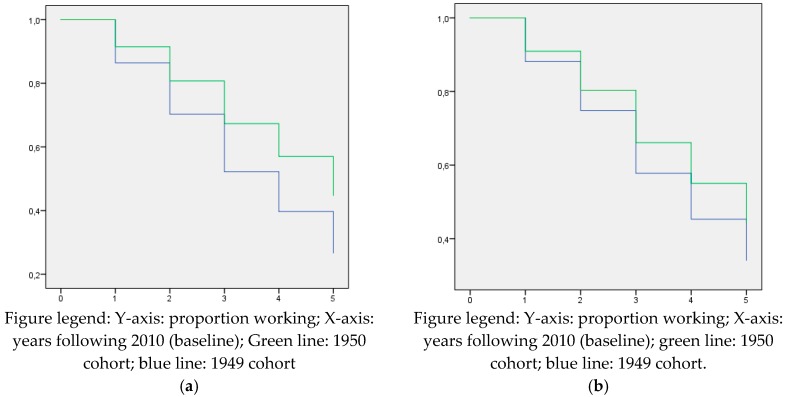
(**a**) Kaplan–Meier curves for male employees for working until the statutory retirement age by the 1949 and 1950 cohorts. (**b**) Kaplan–Meier curves for female employees for working until the statutory retirement age by the 1949 and 1950 cohorts.

**Figure 3 ijerph-16-03895-f003:**
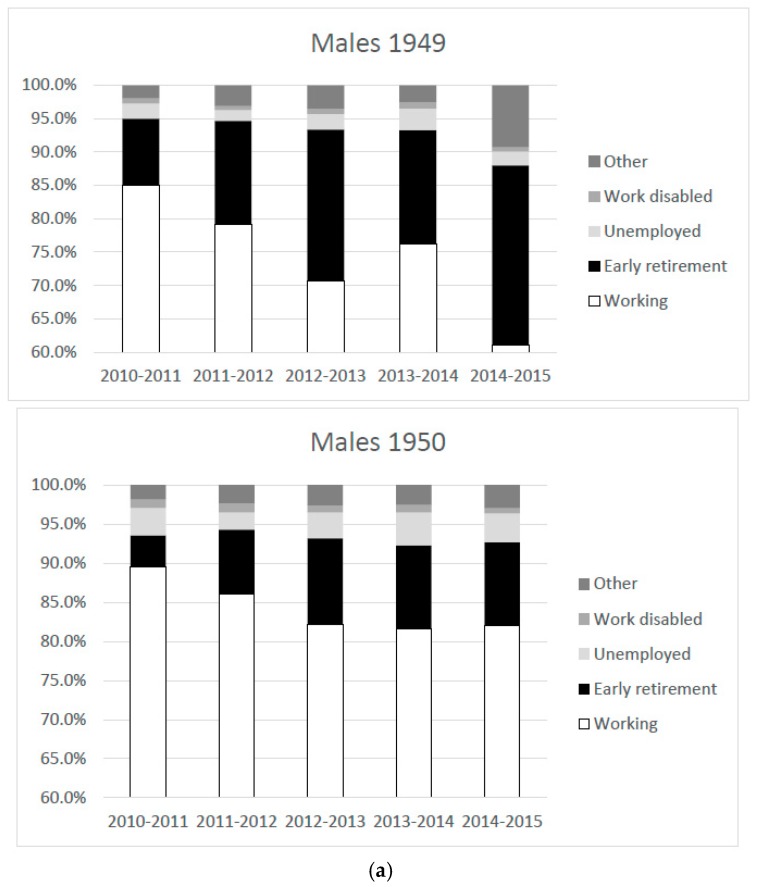
(**a**) Proportions of the top three labor market transitions between 2010 and 2015, in men born in November/December 1949 or January/February 1950, employed in 2010. (**b**) Proportions of the top three labor market transitions between 2010 and 2015, in women born in November/December 1949 or January/February 1950, employed in 2010.

**Figure 4 ijerph-16-03895-f004:**
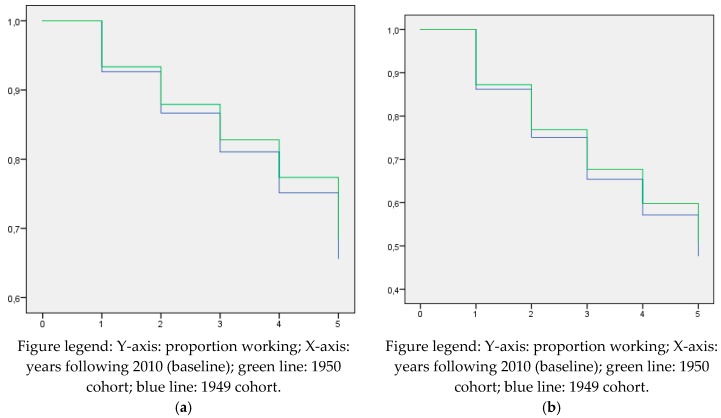
(**a**) Kaplan–Meier curves for male self-employed workers for working until the statutory retirement age by the 1949 and 1950 cohorts. (**b**) Kaplan–Meier curves for female self-employed workers for working until the statutory retirement age by the 1949 and 1950 cohorts.

**Table 1 ijerph-16-03895-t001:** (**a**) Baseline characteristics in 2010 of male employed and self-employed workers born between 1 November 1949 and 28 February 1950. (**b**) Baseline characteristics in 2010 of female employed and self-employed workers born between 1 November 1949 and 28 February 1950.

(**a**)							
	**Total**		**1949 Cohort**		**1950 Cohort**	***p*-Value**	
**Males**	***n***	**%**	***n***	**%**	***n***	**%**	
Total	28,458		13,625		14,833		
Total employed	14,360	50%	6698	49%	7662	52%	<0.001
- Dutch background	12,417	86%	5786	86%	6631	87%	0.78
Self-employed	3367	12%	1607	12%	1760	12%	0.85
- Dutch background	3018	90%	1436	89%	1582	90%	0.62
Total employed + self-employed	17,727	62%	8305	61%	9422	64%	<0.001
- Dutch background	15,435	87%	7222	87%	8213	87%	0.68
(**b**)							
	**Total**		**1949 Cohort**		**1950 Cohort**	***p*-Value**	
**Females**	***n***	**%**	***n***	**%**	***n***	**%**	
Total population	27,767		13,421		14,346		
Employed	9343	34%	4360	32%	4983	35%	<0.001
- Dutch background	8012	86%	3742	86%	4270	86%	0.85
Self-employed	2202	8%	1087	8%	1115	8%	0.31
- Dutch background	1957	89%	957	88%	1000	90%	0.22
Total employed + self-employed	11,545	42%	5447	41%	6098	43%	<0.001
- Dutch background	9969	86%	4699	86%	5279	87%	0.81

**Table 2 ijerph-16-03895-t002:** (**a**) Yearly proportions of male employees born in November of December 1949 or in January or February 1950 that worked until the statutory retirement age or left employment through early retirement, unemployment, or work disability between 2010 and 2015. (**b**) Yearly proportions of female employees born in November or December 1949 or in January or February 1950 that worked until the statutory retirement age or left employment through early retirement, unemployment, or work disability between 2010 and 2015.

(**a**)										
**Males**	**Working**		**Early Retirement**		**Unemployed**		**Work Disabled**		**Other**	
**Cohort**	**1949**	**1950**	**1949**	**1950**	**1949**	**1950**	**1949**	**1950**	**1949**	**1950**
**Years**										
2010–2011	85.0%	89.5%	10.0%	4.1%	2.3%	3.5%	0.8%	1.1%	1.9%	1.8%
2011–2012	79.2%	86.1%	15.5%	8.2%	1.6%	2.2%	0.6%	1.2%	3.1%	2.3%
2012–2013	70.7%	82.2%	22.7%	11.0%	2.3%	3.3%	0.8%	0.9%	3.5%	2.6%
2013–2014	76.2%	81.6%	17.1%	10.7%	3.2%	4.2%	1.0%	1.0%	2.5%	2.5%
2014–2015	61.0%	82.0%	27.0%	10.7%	2.1%	3.7%	0.7%	0.7%	9.2%	2.9%
(**b**)										
**Females**	**Working**		**Early Retirement**		**Unemployed**		**Work Disabled**		**Other**	
**Cohort**	**1949**	**1950**	**1949**	**1950**	**1949**	**1950**	**1949**	**1950**	**1949**	**1950**
**Years**										
2010–2011	88.3%	88.6%	7.8%	6.3%	1.4%	2.3%	0.8%	1.1%	1.7%	1.7%
2011–2012	82.9%	87.1%	12.6%	8.5%	1.9%	2.0%	0.7%	1.0%	1.9%	1.4%
2012–2013	75.0%	81.7%	19.6%	12.0%	2.4%	3.2%	0.9%	1.0%	2.1%	2.1%
2013–2014	77.6%	81.8%	16.9%	12.1%	3.2%	3.3%	0.7%	1.0%	1.6%	1.8%
2014–2015	71.7%	82.2%	23.0%	12.0%	2.0%	3.3%	0.7%	0.5%	2.6%	2.0%

**Table 3 ijerph-16-03895-t003:** Yearly proportions of male and female employees born in November or December 1949 or in January or February 1950 that remained self-employed until the statutory retirement age or left employment between 2010 and 2015.

Self-Employment	Males		Females	
Cohort	1949	1950	1949	1950
Years				
2010–2011	88.4%	90.6%	83.3%	84.3%
2011–2012	90.8%	90.5%	84.2%	84.2%
2012–2013	89.3%	89.8%	82.3%	85.2%
2013–2014	88.7%	90.3%	84.1%	82.5%
2014–2015	82.3%	88.0%	79.3%	83.8%

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
