# Peer review of "Effects of Early Retirement Policy Changes on Working until Retirement: Natural Experiment"

_ijerph, 2019, doi:10.3390/ijerph16203895_

Round 1
Reviewer 1 Report
This is an expertly constructed manuscript by an experienced and highly respected team of researchers. The study described in the manuscript is of considerable importance given the contemporary imperative for longer working lives and ingeniously designed. For me, the final paragraph of the conclusion eloquently and concisely highlights the hugely important potential unintended consequences for workers’ health and wellbeing (and the public purse) of policy initiatives such as that described in this study.
This study is expertly executed. My observations relate to a small set of minor presentational issues:
Page 1, lines 13-14. In the sentence beginning “However, not…” delete the word ‘to’ on its first appearance.
Page 2, lines 54-55. “DI may become more common when ‘treated’ individuals reaching the retirement age are not able to work the extra 13 months because of health problems.” This sentence gives the first mention of a 13-month extension to working life, though the wording implies that the reader is already familiar with the impact of the change in pension rights in terms of the amount of additional working time involved. I’d recommend rewording the opening sentences of this paragraph so as to provide the reader with a clear understanding of the implications of the pension right reform for the number of additional months/ years work required for those born in 1950 or later. To achieve this borrow from the material on lines 98-101.
Page 3, line 94. The title ‘DESIGN’ should be given as ‘Design’ so as to be consistent with earlier sub-headings.
Page 4, line 142. Delete the word ‘a’ on its first appearance.
Page 5, line 193. The first letter of the first word of the sub-heading should be capitalised.
Page 6, lines 195-199. Ensure consistency in number of decimal points to which the percentages are given. Note also that on line 197 a comma rather than decimal point has been given in a percentage.
One observation: I find it astonishing that so little warning – just a matter of months - was given of the change in pension entitlements. In the paragraph at the bottom of page 12 it would perhaps be interesting to inform the international reader of how this was received by the public. This comes at an interesting time given that just last week, on 3rd October 2019, a verdict was announced on a major court case concerning whether women were given sufficient warning of changes in their state pension entitlement. See https://www.bbc.co.uk/news/business-49917315.
Author Response
We would like to thank both reviewers for their constructive comments. Below, we will respond to all points raised by both. In the manuscript, we used track changes to show what we changed.
Reviewer 1
This is an expertly constructed manuscript by an experienced and highly respected team of researchers. The study described in the manuscript is of considerable importance given the contemporary imperative for longer working lives and ingeniously designed. For me, the final paragraph of the conclusion eloquently and concisely highlights the hugely important potential unintended consequences for workers’ health and wellbeing (and the public purse) of policy initiatives such as that described in this study.
This study is expertly executed. My observations relate to a small set of minor presentational issues:
Page 1, lines 13-14. In the sentence beginning “However, not…” delete the word ‘to’ on its first appearance.
Response: Thank you, we have now revised the sentence accordingly.
Page 2, lines 54-55. “DI may become more common when ‘treated’ individuals reaching the retirement age are not able to work the extra 13 months because of health problems.” This sentence gives the first mention of a 13-month extension to working life, though the wording implies that the reader is already familiar with the impact of the change in pension rights in terms of the amount of additional working time involved. I’d recommend rewording the opening sentences of this paragraph so as to provide the reader with a clear understanding of the implications of the pension right reform for the number of additional months/ years work required for those born in 1950 or later. To achieve this borrow from the material on lines 98-101.
Response: We have now added the following sentence as introduction for the pension reform (page 2, lines 51-53):
The group born in 1950 or later either had to work another 13 months to obtain the same pension rights, or would have to accept a pension of 64% instead of 70% of their gross wages if they wanted to retire at age 62 years and 3 months.
Page 3, line 94. The title ‘DESIGN’ should be given as ‘Design’ so as to be consistent with earlier sub-headings. Page 4, line 142. Delete the word ‘a’ on its first appearance. Page 5, line 193. The first letter of the first word of the sub-heading should be capitalised. Page 6, lines 195-199. Ensure consistency in number of decimal points to which the percentages are given. Note also that on line 197 a comma rather than decimal point has been given in a percentage.
Response: thank you for notifying these 4 issues, we have now changed all accordingly.
One observation: I find it astonishing that so little warning – just a matter of months - was given of the change in pension entitlements. In the paragraph at the bottom of page 12 it would perhaps be interesting to inform the international reader of how this was received by the public. This comes at an interesting time given that just last week, on 3rd October 2019, a verdict was announced on a major court case concerning whether women were given sufficient warning of changes in their state pension entitlement. See https://www.bbc.co.uk/news/business-49917315.
Response: It was indeed astonishing that the implementation of the reform was just a few months after it was announced to the public. This was the topic of a paper by De grip, Lindeboom and Montizaan examined the impact of the reform on the mental health of workers.
De Grip, Lindeboom, Montizaan. Shattered dreams: the effects of changing the pension system late in the game. The Economic Journal, 122 (March), 1–25. Doi: 10.1111/j.1468-0297.2011.02486.x.
As our paper focuses on the effects of labour participation, we chose not to include it in the manuscript. For your information, this is some information about the effects on mental health:
They write in their paper:
"In the light of demographic changes, it had been acknowledged [..] that reform of the pension system would be necessary. In that sense, a change in pension rights was not entirely unexpected. However, the timing of the reform as well as the particular implementation of a discontinuous assignment rule and the strong differential treatment of workers born around 1 January 1950 came as a surprise [......] when it was announced on 5 July 2005."
They find strong negative effects on the mental health of affected workers born in 1950:
“We find that the reduction in pension rights is important for mental health but that other factors are also at work. The discontinuous assignment rule and the strong differential treatment of workers born around 1 January 1950 is likely perceived to be unfair and may have led to severe disappointment. Moreover, the pension reform was announced only a few years before the retirement date of the affected workers and too little time remained to allow these workers to fully offset the loss in pension wealth.
Workers were suddenly forced into a new situation, with little control over their retirement decision which may have affected their mental health."
And state (also not to be included in our manuscript):
"The results of this study show that a sudden irreversible deterioration of future prospects can have serious consequences for the mental health of workers nearing retirement, especially when their own employer reneges on pre-existing arrangements (violates an implicit
contract) in ways that are difficult to adjust to once one has taken those rules into account in one’s plans. The period before the planned retirement is too short to compensate for losses in pension wealth."
Reviewer 2 Report
This is a very nice paper with a straightforward research question and very good analysis. It is written in good English and the conclusion are well draw. From my side only four comments.
One “larger” one:
I would recommend to be a bit more “careful” regarding the conclusion on the increase in the unemployed and disability spell. Of course there is an increase from the 1949 to the 1950 cohort, but I am not too convinced that it is a substantial as interpreted by the authors. The main change is from early retirement to employment. One additional sentence that the increase is from a rather low level would be enough.Three “smaller” comments:
On page 6 in line 197: I think it should be: For the treated the employment rates are substantially higher In Figure 3 I would recommend to use a different color for the category early retirement. The black lines are very confusing. Maybe a light grey. On page 1 in line 34 it says: Economic factors play an important role… As a sociologist (I assume the authors are economist) I am not too sure if this applies to all groups. For the low income group this might be the case. I make a huge different if you have a pension of 1000 or 1500 Euros. The difference should be less important between 4500 and 5000 Euros. So I would recommend deleting the important. However, I would also not be nitpicking and I am also happy if the authors decide to stay with the important.Author Response
We would like to thank both reviewers for their constructive comments. Below, we will respond to all points raised by both. In the manuscript, we used track changes to show what we changed.
This is a very nice paper with a straightforward research question and very good analysis. It is written in good English and the conclusion are well draw. From my side only four comments.
One “larger” one:
I would recommend to be a bit more “careful” regarding the conclusion on the increase in the unemployed and disability spell. Of course there is an increase from the 1949 to the 1950 cohort, but I am not too convinced that it is a substantial as interpreted by the authors. The main change is from early retirement to employment. One additional sentence that the increase is from a rather low level would be enough.
Response:
Three “smaller” comments:
On page 6 in line 197: I think it should be: For the treated the employment rates are substantially higherResponse: Thank you for noticing this error, we have now corrected this.
In Figure 3 I would recommend to use a different color for the category early retirement. The black lines are very confusing. Maybe a light grey.Response: We have now changed the colours in the graph.
On page 1 in line 34 it says: Economic factors play an important role… As a sociologist (I assume the authors are economist) I am not too sure if this applies to all groups. For the low income group this might be the case. I make a huge different if you have a pension of 1000 or 1500 Euros. The difference should be less important between 4500 and 5000 Euros. So I would recommend deleting the important. However, I would also not be nitpicking and I am also happy if the authors decide to stay with the important.Response: We have now removed ‘important’ from this sentence.